# Comparing Primary and Secondary Growth of Co-Occurring Deciduous and Evergreen Conifers in an Alpine Habitat

**Yiping Zhang [1], Yuan Jiang [2],\*, Yan Wen [2], Xinyuan Ding [2], Biao Wang [2] and Junliang Xu [1]**

[1]  College of Forestry, He'nan University of Science and Technology, Luoyang 471003, China
[2]  Beijing Key Laboratory of Traditional Chinese Medicine Protection and Utilization, Faculty of Geographical Science, Beijing Normal University, Beijing 100875, China
\*  Correspondence: jiangy@bnu.edu.cn; Tel.: +86-010-5880-9274

**Abstract:** Investigations on primary and secondary growth in co-occurring species will aid in assessment of the physiological adaptation of species and the prediction of forest stand structure dynamics. To explore the correlation and coordination between primary and secondary growth, we monitored the leaf phenology, shoot elongation, and stem growth of co-occurring *Larix principis-rupprechtii* Mayr. and *Picea meyeri* Rehd. et Wils. in an alpine habitat, Luya Mountain (North-Central China), during the growing season of 2014. We measured bud development on terminal branches three days per week by direct observations and intra-annual stem xylem formation at weekly intervals by the microcores method. The onset sequence of three organs was the needle, shoot, and stem, without species-specific differences. Needles appeared one month earlier than stem growth in larch, while it was only one week earlier in spruce. The duration of needle growth was the shortest, followed by the shoot and stem. The timing of primary growth (i.e., onset, end, and maximum growth rate) between the two species was asynchronous, but secondary growth was synchronic with the same date of the maximum growth rate occurrence, potentially indicating species competition for resources. Unlike larch, spruce staggered growth peaks among different organs, which may effectively mitigate trees' internal competition for resources. Soil temperature was positively correlated with both shoot and stem growth in the two species, whereas air temperature and soil water content were positively correlated with needle growth only in larch. Therefore, it can be inferred that the spruce will probably outcompete the larch at cold alpine treeline sites due to its high adaptability to acquiring and allocating resources. These results provide insight into the potential physiological correlation between primary and secondary growth and allow better prediction of future climate change effects on forest ecosystem productivity.

**Keywords:** budburst; xylogenesis; phenology; tree growth; soil temperature

## 1. Introduction

Primary (e.g., budburst, needle, and shoot) growth and secondary growth (e.g., cambial activity and xylem differentiation) of plants play a vital role in long-term carbon sequestration in terrestrial ecosystems [1–3]. Based on empirical phenological observations, earlier bud burst, leafing, shooting, and flowering have been recorded for these sequential phases because of the recent early spring warming and extended growing season length [4]. Studies on secondary growth have revealed an early onset of cambium activity, as well as a longer duration of xylem formation, and this has often been related to the influences of ongoing climate change [5,6]. Therefore, investigations on primary and secondary growth help to assess the physiological adaptation of species to the local environment [7] and predict changes in the growing season under current global warming [8].

Recently, the correlation and coordination between primary and secondary growth have received much attention. Huang et al. [9] quantified that 74% of stem formation could be explained by primary growth. Antonucci et al. [7] and Perrin et al. [10] further proved there were strong temporal relationships between the beginning of bud development and reactivation of xylem stem growth in early spring. There are time lags between leaf and stem growth, leading to differences in carbon allocation, which could elucidate the physiological relationships between meristems [11]. In ring-porous trees, leaf budburst generally occurs following stem growth [12,13]. Early stem growth is associated with winter embolism of large xylem vessels, which requires that the restoration of the water flow pathway occurs each spring before the onset of transpiration [14]. However, for diffuse-porous and coniferous species, leaf budburst may start before, after, or be synchronized with stem growth, which depends on the species specifics and local environment [15–17]. Zhai et al. [18] investigated two co-occurring diffuse-porous trees and found that budburst began in *Betula papyrifera* Marsh. before stem formation but in *Populus tremuloides* Michx. after stem formation. Moser et al. [19] reported that needles in *Larix decidua* Mill. appeared several weeks before the onset of stem growth occurred in the central Swiss Alps. In contrast, Rossi et al. [17] reported that needle and shoot growth of *L. decidua* occurred after reactivation of stem growth in the eastern Italian Alps. Hence, these contrasting results indicate that relationships between primary and secondary growth are not constant but may vary among tree species and could change under local environmental conditions, which need further clarification.

Prince Rupprecht's larch (*Larix principis-rupprechtii* Mayr.) and Meyer's spruce (*Picea meyeri* Rehd. et Wils.) are two dominant species within cold temperature coniferous forests (1800–2800 m a.s.l.) in North-Central China [20]. These two co-occurring conifers show different successional and phenological traits. Unlike evergreen spruce, whose long-lived needles can provide a potentially longer photosynthetic season, larch rebuilds its tree canopy several weeks earlier to be capable of photosynthesis in spring [21]. Thus, deciduous larch, which prefers light habitats, has been considered as an early-successional species, while evergreen spruce prefers to grow in shady habitats as a late-successional species [22]. It has been theorized that early-successional species are more prone to take risks, while late-successional species are associated with safer life strategies [23]. An early onset of growth may be linked to high risks, e.g., increasing exposure to spring frosts [24]. Cuny et al. [25] investigated the life strategies of three co-occurring conifer species for intra-annual stem formation and found that pioneer species (pines) are greater resource expenders and develop riskier life strategies to capture resources, while shade-tolerant species (firs and spruces) utilize resources more efficiently and develop safer life strategies. The differences observed between early and late-successional species in budburst [26] and cambial activity resumption [27] may provide a better estimate of which species would be more vulnerable to future climate change.

There are still limited ecophysiological studies that focus on explaining the relationships between primary and secondary growth on co-occurring trees [9,28]. The objectives of this study were (1) to compare the timing sequences and growth rate between primary (needle and shoot) and secondary growth (stem) in two co-occurring conifers (larch and spruce) during a growing season, and (2) to analyze the potential relationships between the growth of different organs and environmental variables. Previous studies found that the temperature, either threshold value, or accumulated value (degree-days) is crucial to understanding the differentiation of meristem cells and their coordination. We hypothesized that (1) the timing of primary and secondary growth (i.e., onset, end, and maximum growth rate) is different and coordinated due to internal competition for resources among organs, and (2) the growth (both primary and secondary) for larch and spruce under the same growing conditions is species-specific and asynchronous because of their different life forms and limited resource availability in cold alpine conditions.

## 2. Materials and Methods

### 2.1. Study Site and Species Descriptions

The study site is the treeline ecotone situated on the summit of Luya Mountain (2740 m a.s.l.), North-Central China (38°40′ N–38°50′ N, 111°50′ E–112°00′ E, Figure 1a). The climate is semi-humid temperate with monsoon rainfall during summer. The mean annual temperature and precipitation at the base of the mountain are approximately 5.2 °C and 473 mm (1957–2013). The year of the study (2014) was a wet year, with a 20.4% higher annual precipitation than the long-term average (1957–2013). However, this surplus precipitation mainly fell in July as 192.6 mm of rain, which was 70.4% higher than the long-term average for the same period (1957–2013, Figure 2).

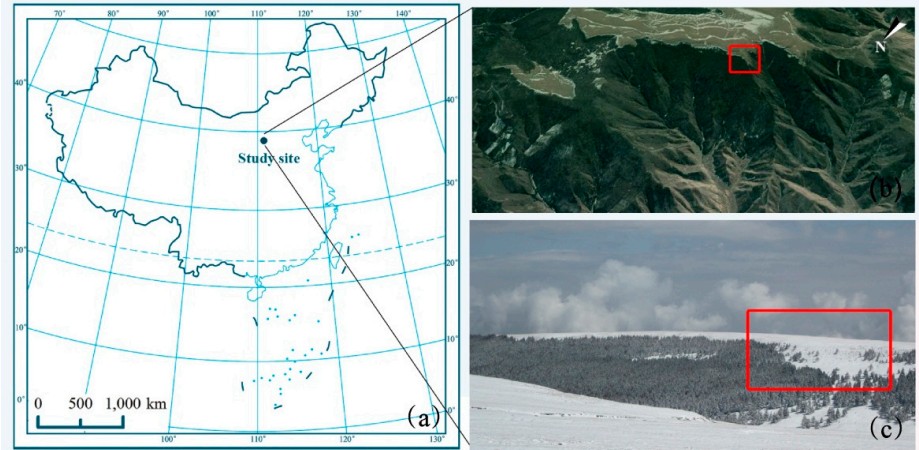

**Figure 1.** (**a**) Location of the study area in North-Central China; the study area landscape in summer (**b**) and winter (**c**).

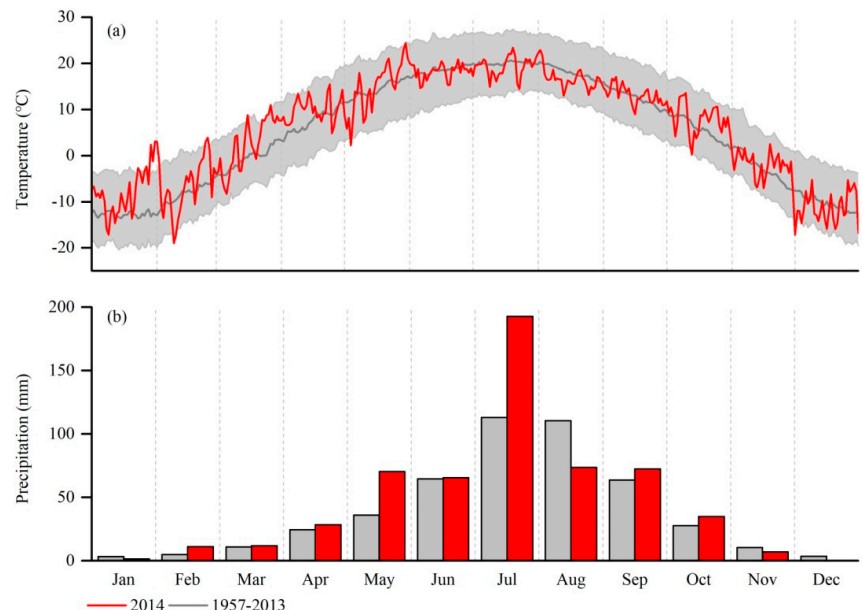

**Figure 2.** Comparison of (**a**) the mean daily air temperatures and (**b**) the monthly sums of precipitation for the monitoring year of 2014 (red) and the 1957–2013 average (light grey). The grey shading in (**a**) indicates the mean minimum and maximum temperatures averaged over the 1957–2013 period. Data were obtained from the nearest state standard meteorological station in Wuzhai (38°55′ N, 111°49′ E, 1401 m a.s.l., linear distance approximately 20 km north of the study site).

*L. principis-rupprechtii* and *P. meyeri* are two dominant species. These trees are very sparse and isolated, as the study site is near the treeline ecotone (Figure 1b,c), which is in the transition zone from cold coniferous forest (1850–2700 m a.s.l.) to subalpine meadow (2700–2780 m a.s.l.) [29]. Primary and secondary growth were monitored over 7 months from April to October 2014, in these two species. To determine the onset of phenology accurately, at the beginning of sample processing, the sampling frequency was at intervals of 3 to 5 days. Once the full elongation of needles (95% of needles had finished 95% length growth) in trees finished at the end of June, sampling was set at 7 to 10 day intervals. Five trees per species were randomly selected in each site, with healthy dominant or co-dominant individuals chosen (Table 1).

**Table 1.** Mean (standard deviation) of the characteristics of the sample trees.

| Species | Height (m) | Diameter (cm) | Age (year) | Crown Breadth (m × m) |
|---|---|---|---|---|
| *Larix principis-rupprechtii* Mayr. | 7.8 (1.8) | 21.1 (2.9) | 57 (6) | 3 × 3 |
| *Picea meyeri* Rehd. et Wils. | 8.5 (1.5) | 21.5 (4.6) | 58 (9) | 4 × 3 |

Age and diameter recorded at breast height.

### 2.2. Primary Growth

Two north-facing and two south-facing branches per tree were selected in the bottom part of the canopy. On each branch, the phases of bud development were recorded on terminal buds. Due to the phenological differences between deciduous and evergreen conifers, their primary phenological phases were distinguished according to Migliavacca et al. [30] for larch and Huang et al. [9] for spruce as follows (Figure 3): (1) Bud burst: Green needle tissue visible but needle length <1 cm for larch and upper part of the bud with smooth and pale-colored scales but no visible needles for spruce. (2) Needles unfolding: Needles were fully expanded (>3 cm) and elongated for larch. For spruce, this stage was hard to distinguish because, at this time, needles were still covered by bud scales, while the tops of the needles apparently grew [25]. (3) New shoot lengthening: New shoot has undergone elongation and finished 5% of length growth. Shoot elongation was determined on long-shoots for both larch and spruce. (4) Needles stop elongation: 95% of needles finished 95% length growth. (5) New shoots stop lengthening: 95% of shoots finished 95% length growth. (6) Leaf senescence: 95% of larch needles turned from green to yellow. (7) Leaf defoliation: 95% of larch needles turned from yellow to bronze. The lengths of the needle and shoot elongation were measured by a steel ruler with a 1 mm precision. All data were computed in days of the year (DOY) by averaging the data observed in the field. The duration of needle growth was calculated as the number of days between bud burst and the end of needle elongation. The duration of shoot growth was defined as the number of days between shoot elongation and the end of shoot elongation.

### 2.3. Secondary Growth

Secondary growth can be determined by monitoring stem cambial activity and xylem formation using the microcore method on a weekly or biweekly basis. This microcore method is an effective technique that has been widely accepted and used in dendroecology [1]. Secondary growth was monitored on the same dates and in the same trees as the bud phenology was measured. Each time, five microcores per tree were collected from stems at breast height (1.3 m) using a Trephor tool [31]. These microcores were then prepared in the laboratory and cut using a microtome to obtain a thin slice (6–10 μm thickness). Each slice was stained with 1% safranin and 0.5% fast green (in 95% ethanol) to identify the different development phases of cambium cells and xylem cells (Figure 4) [32].

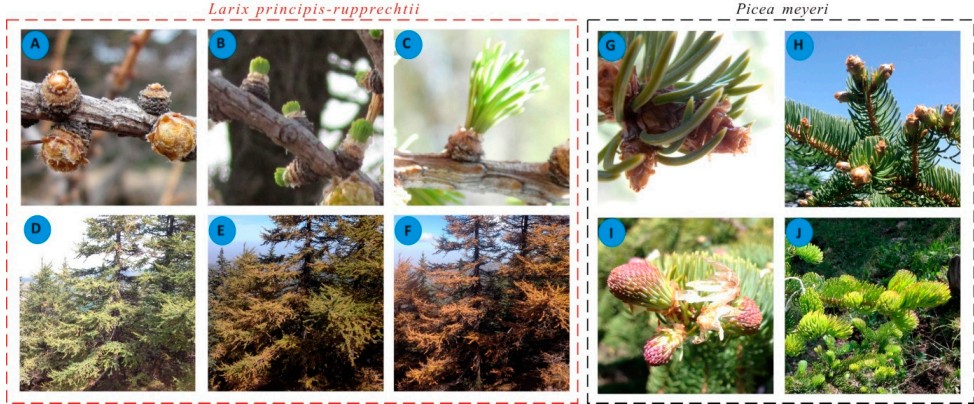

**Figure 3.** Illustrations of some stages of primary growth in *Larix principis-rupprechtii* (**A–F**) and *Picea meyeri* (**G–J**). **A**: unexpanded bud (18 April); **B**: bud burst (5 May); **C**: needles unfolding (16 May); **D** and **E**: leaf senescence (18 September and 25 September, respectively); **F**: defoliation (2 October) for larch; **G**: bud burst (25 May); **H**: new shoot elongation (18 April); **I**: needles unfolding (2 June); **J**: new shoot growth (21 June) for spruce.

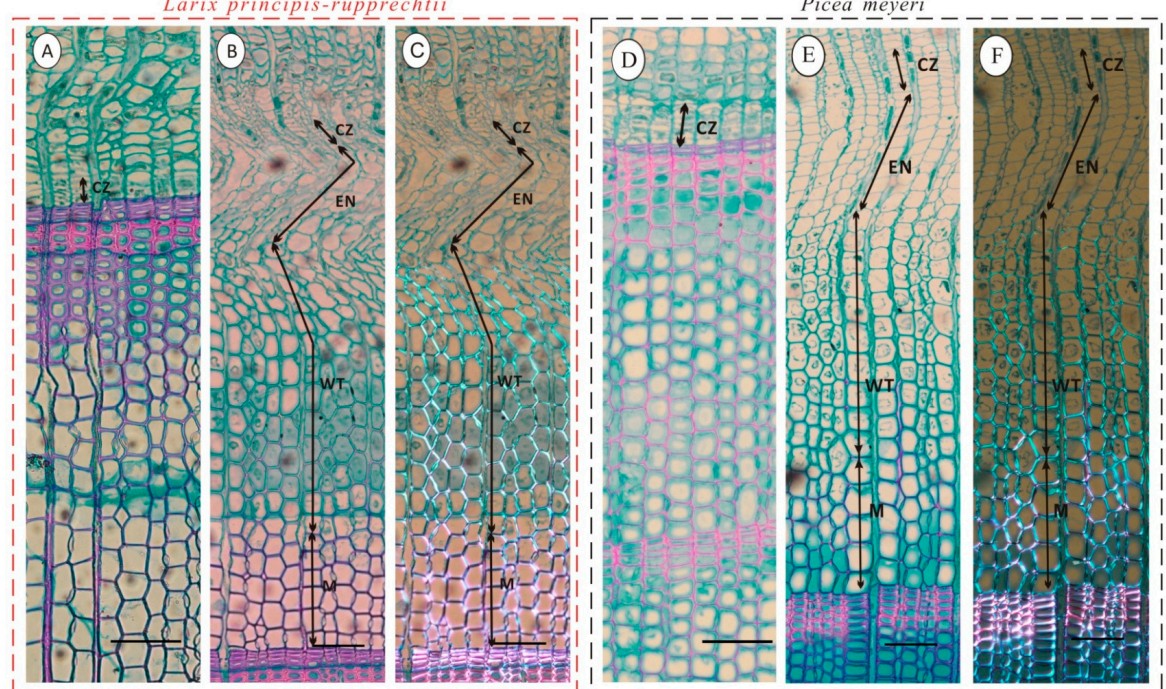

**Figure 4.** Phases of secondary growth in *Larix principis-rupprechtii* (**A–C**) and *Picea meyeri* (**D–F**) on the same date. (**A,D**): on 24 April, no new xylem cell production was observed and the cambial zone (CZ) consisted of seven to eight cells and five to six cells for larch and spruce, respectively; (**B,E**) and (**C,F**): on 20 July, enlarging (EN), wall-thickening (WT), and mature (M) cells under bright field and polarized light, respectively; scale bar = 100 μm.

Secondary growth (cambial activity and xylem differentiation) was divided into the cambial zone (CZ), radial enlargement phase (EN), cell wall thickening phase (WT), and mature phase (M). Cambial cells had small radial diameters and thin cell walls. Enlargement cells were characterized by thin primary cell walls but with a radial diameter at least two times that of a cambial cell. Polarized light was used to discriminate between enlargement and wall thickening tracheids. Mature cells had empty cell lumens and red cell walls. Therefore, the secondary growth stages were defined as: (1) $CZ_{onset}$, (2) $EN_{onset}$, (3) $WT_{onset}$, (4) $M_{onset}$, (5) $CZ_{end}$, (6) $EN_{end}$, and (7) $WT_{end}$. In spring, cambial cells starting to increase after dormancy was defined as $CZ_{onset}$. The first appearance in spring of cells

in the enlargement phase was defined as $EN_{onset}$. The end of cell wall thickening and lignification in late summer was defined as $WT_{end}$. The duration of xylogenesis was calculated as the number of days between $EN_{onset}$ and $WT_{end}$.

### 2.4. Weather Station

In the study site (2740 m a.s.l.), an automatic weather station was installed in a relatively flat and unshielded area to measure environmental factors. The air temperature probe (HOBO Pro V2, Onset Computer Corp., Bourne, MA, USA) was fixed at a height of 1.5 m above the ground and the soil temperature probe (ST-05, Delta-T, Cambridge, UK) was buried at a depth of 10 to 20 cm. The soil volumetric water content (SWC) was observed (with a PR2 instrument, DeltaT, City, UK) at a depth of 10 to 20 cm. Precipitation was also measured (with Davis 7852 instrument, Hayward, CA, USA). All these environmental data were recorded at 30 min intervals by CR10X data loggers (Campbell Scientific Corporation, Logan, UT, USA).

### 2.5. Data Analysis

The differences in the timing and duration of primary and secondary growth between the two species were determined using the Mann–Whitney rank-sum test. The phenological date differences (onset, end, and maximum growth rate) among the three organs were evaluated with the Kruskal–Wallis test due to a lack of normality and homoscedasticity of the data, and an ANOVA could not be conducted [33]. Subsequently, a multiple comparison test after Kruskal–Wallis ($a = 0.05$) was performed with SPSS (SPSS Inc., Chicago, IL, USA).

To assess the growth dynamics, the length of needles (shoots) and the total number of xylem cells for primary and secondary growth, respectively, were modeled for the two species with the Gompertz function, which has been widely accepted to describe intra-annual growth of needles, shoots, and stems [18].

The relationships of primary (needle, shoot)–secondary growth and growth–environmental factors were analyzed using Pearson correlation coefficients. Considering the comparison, data were also converted in the form of the percent of final growth. Growth increases were calculated for sample intervals based on the Gompertz function. For the same interval, the daily mean air and soil temperature, soil volumetric water content, precipitation sums, relative humidity, and growing degree days of the mean air temperature ($GDD_a$) and mean soil temperature ($GDD_s$) (presented in Supplementary Materials S1) were also calculated.

## 3. Results

### 3.1. Primary Growth

Larch started primary growth (needle and shoot growth) earlier than spruce (Figure 5a,b), with the onset and end timing of each phase between the two species having significant differences (Mann–Whitney rank-sum test, $p < 0.05$, Table S1). The occurrence of bud burst was on DOY 125 (5 May) and 145 (25 May) in larch and spruce, respectively, and the following bud phenology was subsequently earlier in larch than in spruce. Needle growth was more than one month and lasted 46 days for larch from early May to mid-June and 31 days for spruce, mainly in June. Shoot elongation occurred approximately three and two weeks after bud burst in larch and spruce, respectively. Shoot growth duration, which was much longer in larch than in spruce, was 59 days (DOY 142–201, 22 May –20 July) and 33 days (DOY 153–186, 2 June–5 July), respectively.

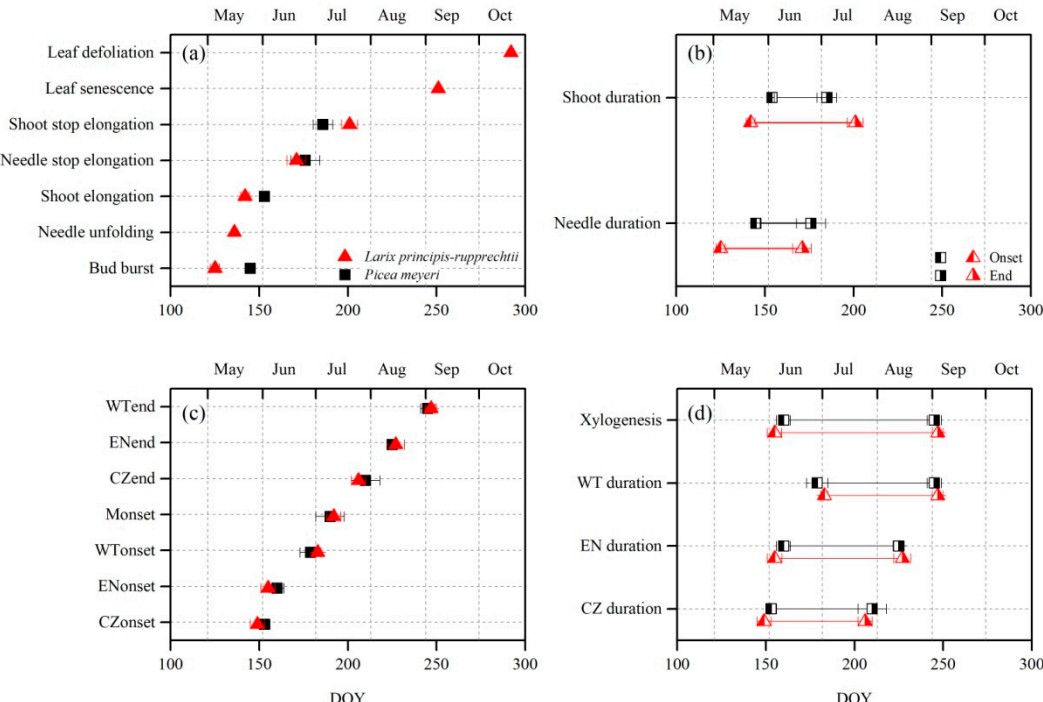

**Figure 5.** Dates of occurrence of primary (**a**,**b**) and secondary growth (**c**,**d**) for spruce and larch. (CZ: cambial zone, EN: enlargement cell, WT: wall thickening cell, M: mature cell). Error bars indicate the standard deviation between trees.

### 3.2. Secondary Growth

Larch and spruce had similar trends of stem xylem formation in the study year, with no significant differences in the onset and end timing of each phase between the two species (Figure 5c,d, Table S2). Cambial resumption occurred in late May and the beginning of June; approximately one week later, the first enlarging cells were formed. One month later, the first mature cells appeared after the start of enlargement. The termination of cambial activity and xylem formation were recorded in late July and the beginning of September, respectively. Therefore, the duration of stem xylem formation was 92 days and 85 days for larch and spruce, respectively.

### 3.3. Comparing Primary and Secondary Growth

Primary growth preceded secondary growth in both larch and spruce ($p < 0.05$, Table 2, Figure 6). The needle growth of larch started one month earlier than stem growth due to its deciduousness, while it was only one week earlier for spruce. Not only the onset timing between primary growth and secondary growth but also the end timing significantly differed. Therefore, for the duration of growth, needle growth was shortest, followed by the shoot and stem.

**Table 2.** Comparison of dates (onset, end, and maximum growth rate) among three organs in larch and spruce (mean ± standard deviations).

| Species | Organ | Onset (DOY) | χ (p) | End (DOY) | χ (p) | $t_p$ (DOY) | χ (p) |
|---------|-------|-------------|-------|-----------|-------|-------------|-------|
| Larch | Needle | 125 ± 1 [aA] | 8.889 (0.012) | 171 ± 2 [aA] | 9.143 (0.010) | 150 ± 3 [aA] | 5.778 (0.056) |
| | Shoot | 142 ± 2 [bA] | | 201 ± 4 [bA] | | 185 ± 1 [bA] | |
| | Stem | 155 ± 4 [cA] | | 247 ± 3 [cA] | | 183 ± 3 [bA] | |
| Spruce | Needle | 145 ± 2 [aB] | 8.848 (0.012) | 176 ± 4 [aB] | 8.434 (0.015) | 161 ± 5 [aB] | 8.727 (0.013) |
| | Shoot | 153 ± 1 [bB] | | 186 ± 6 [aB] | | 169 ± 2 [aB] | |
| | Stem | 160 ± 4 [cA] | | 245 ± 4 [bA] | | 183 ± 5 [bA] | |

Larch = *Larix principis-rupprechtii*; spruce = *Picea meyeri*; DOY = day of the year, $t_p$ = the date of the maximum growth rate occurrence (the inflection point); lower letters (a,b) indicate significant differences among three organs based on the Kruskal–Wallis Test ($p < 0.05$) and upper letters (A,B) indicate significant differences between two species based on the Mann–Whitney rank-sum test ($p < 0.05$).

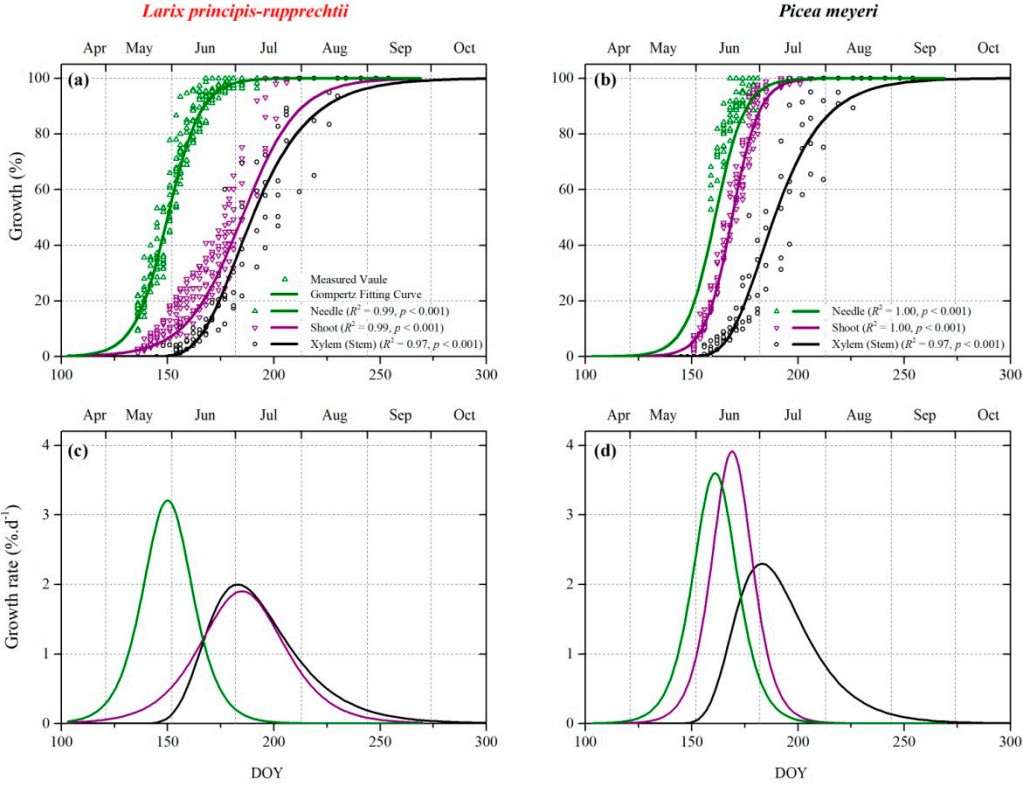

**Figure 6.** (**a**,**b**) Intra-annual dynamics of primary growth and secondary growth and (**c**,**d**) the daily growth rate of larch and spruce. Measured data (scatter) are given as a percent of the final growth and modeled by the Gompertz function (line) (for parameters, see Table S3).

The growth rates of all organs showed a normal curve during the growing season and peaked from the end of May to the end of June (Figure 6). Overall, the fastest rate with the earliest peak was found for needle growth compared with shoot and stem growth. For larch, the occurrence of maximum growth rates of the shoot and stem were roughly synchronized (Table 2). However, for spruce, the maximum growth rates of the three organs distinctly differed and appeared sequentially in the order of needle, shoot, and stem (Figure 7). In addition, the maximum stem growth rate for both species occurred on the same day (DOY 183, 2 July, Figure 6c, Figure 7) and with similar values.

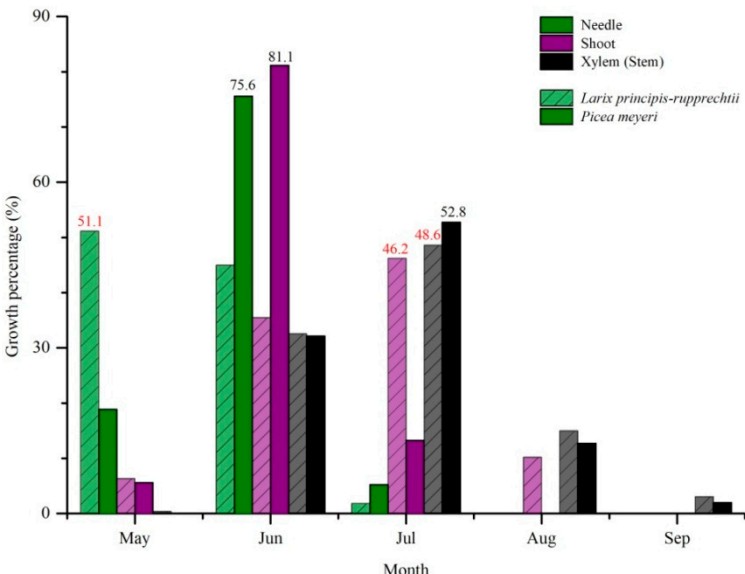

**Figure 7.** Percentages of primary and secondary growth at monthly intervals for larch and spruce.

The correlations between primary and secondary growth were highly significant (Table 3). A strong association was detected between shoot growth and stem growth for both larch ($r = 0.969$, $p < 0.001$) and spruce ($r = 0.414$, $p < 0.01$). Needle growth and stem growth showed a significant correlation in larch ($r = -0.336$, $p < 0.05$) but not in spruce ($r = -0.167$, $p > 0.05$), meaning that the variation between both stages is not always synchronous for spruce. Moreover, needle growth and shoot growth showed a significant correlation only in spruce ($r = 0.561$, $p < 0.01$).

**Table 3.** Correlation matrix between primary (needle, shoot) and secondary growth (stem) for larch (white background) and spruce (grey background).

|  | **Needle** | **Shoot** | **Stem** |
|---|---|---|---|
| Needle | - | −0.238 | −0.336 [*] |
| Shoot | 0.561 [***] | - | 0.969 [***] |
| Stem | −0.167 | 0.414 [**] | - |

A significant correlation is marked with star, and *, **, and *** correspond to $p < 0.05$, $<0.01$, and $<0.001$, respectively.

### 3.4. Environment–Growth Relationship

Soil temperature, especially the degree-days sum of soil temperature, was significantly and positively correlated with both shoot growth and stem growth for the two species but not with needle growth (Table 4). The daily air temperature and SWC were positively correlated with needle growth in larch ($r_{Ta} = 0.816$, $r_{SWC} = 0.769$, $p < 0.05$) but not in spruce.

**Table 4.** Pearson correlation coefficients between tree growth (primary and secondary) and environmental factors of larch and spruce.

| Organ | Species | Ta | Ts | SWC | P | RH | GDD$_a$ | GDD$_s$ |
|---|---|---|---|---|---|---|---|---|
| Needle growth | Larch | 0.816 * | 0.661 | 0.769 * | 0.210 | 0.047 | 0.610 | 0.681 |
| | Spruce | 0.796 | 0.633 | −0.771 | −0.765 | −0.137 | 0.795 | 0.827 |
| Shoot growth | Larch | 0.312 | 0.946 ** | −0.07 | 0.445 | 0.454 | 0.621 * | 0.985 *** |
| | Spruce | 0.467 | 0.841 * | −0.273 | 0.008 | -0.582 | 0.575 | 0.938 ** |
| Stem growth | Larch | 0.636 | 0.931 ** | −0.016 | 0.427 | 0.158 | 0.874 * | 0.958 ** |
| | Spruce | 0.335 | 0.904 ** | 0.590 | 0.543 | 0.670 | 0.746 | 0.955 ** |

Larch = *Larix principis-rupprechtii*; spruce = *Picea meyeri*; Ta = mean air temperature, Ts = mean soil temperature, SWC = soil volumetric water content, P = precipitation, RH = relative humidity, GDD$_a$, GDD$_s$ = growing degree days of mean air temperature and mean soil temperature; significant correlation is marked with star, and *, **, and *** correspond to $p < 0.05$, <0.01, and <0.001, respectively.

## 4. Discussion

The results supported the first hypothesis that the timing (i.e., onset, end, and maximum growth rate) between primary and secondary growth is significantly different ($p < 0.05$, Figure 5, Table 2, Table S3) for both larch and spruce, except for the maximum growth rate between the shoot and stem for larch. However, the second hypothesis that there were species differences in growth was only partially accepted (Figure 6, Table 3) because species-specific asynchrony only occurred in primary growth (needle and shoot) but not in secondary growth (stem xylem).

### 4.1. Coordination and Correlation between Primary and Secondary Growth

Our results showed that the onset sequence of the three organs was needle, shoot, and stem without species-specific differences. For larch, needles appeared approximately two weeks and four weeks earlier than shoot and stem growth, respectively (Figure 5, Table 3), while for spruce, it was only one week and two weeks earlier, respectively. This result is consistent with the finding of Moser et al. [19], who reported that needles in *L. decidua* appeared approximately three to four weeks before the onset of radial stem growth occurred. Similarly, Swidrak et al. [28] found needle growth in co-occurring *L. decidua* and *P. abies* was nearly completed before and simultaneously, respectively, with the onset of stem growth. This finding is in agreement with the hypothesis that the formation of early xylem cells might be triggered by auxin in the newly developing leaves and shoots during early growth [34]. Auxin is essential for wood formation and may be transported to stimulate cambium reactivation [35]. In contrast, it also has been reported that secondary stem growth may start earlier than primary growth [17]. This discrepancy could be explained by genetic differences related to the species-specific and environmental differences related to the particular site [3].

Consistent with the onset timing sequence, the end sequence of the three organs was also needle, shoot, and stem. Additionally, the mean growth rate (relative percent) of the needle, shoot, and stem were 1.94%, 1.20%, and 1.22% and 2.02%, 2.33%, and 1.43% for larch and spruce, respectively (Table S3). Therefore, the duration was shortest in needle growth and longest in stem growth. This result is owing to the plant carbon acquisition via leaf photosynthesis, with the leaf having a higher priority to uptake and allocate carbon than the stem [36]. Martínez-Vilalta et al. [37] assembled a global database to examine patterns of seasonal NSC variation across organs and pointed out that NSC concentrations were highest in leaves and lowest in stems, which was consistent with our results. Shoots play a connective role between the carbon–source organ (leaf) and the carbon–sink organ (stem) [38]. After trees finished canopy building, surplus carbohydrates were transferred by shoots, and then invested into stem growth and/or root storage; this phenomenon explained the time lag between the needle growth and stem xylem growth [39].

### 4.2. Synchronisms of Secondary Growth but Asynchronisms of Primary Growth between Species

Primary phenological differences were observed between the two species (Figure 5, Table 2). Larch began budburst approximately 20 days earlier than spruce. These behaviors could be explained by the different life forms of theirs leaves, related to the species-specific capacity to start leaf photosynthesis during the early growing season. As deciduous species, larch need to guarantee enough time for foliar canopy development, otherwise its growth would be relative constrained. Consistent with our finding, phenological observations based on digital images have also found earlier green-up and later senescence of *Larix kaempferi* in the mixed forest in the highlands of Japan [40].

However, secondary growth between larch and spruce was synchronic, with no significant differences in both the onset and end timings and the occurrence of the maximum growth rate (Figure 6). These synchronisms observed between the two species could be the result of functional convergence of the trees [41] with similar responses of stem growth to a common driving factor (e.g., photoperiod may constraint the maximum stem growth rate to occur close to summer solstice). Moreover, this synchronicity of stem xylem growth also suggests species competition for resources [25]. It has been pointed that trees in a mixed forest tend to reduce competition in two dimensions: First, in space, e.g., by the complementarity of crown and root architectures; second, in time, by the occupation of different time windows to carry out different vital processes [42]. In our study, the maximum growth rate of xylem growth peaked at the same time for two species, indicating that most wood formation occurred at the same time for all trees, which might aggravate tree-to-tree competition in cold alpine locations.

Compared with spruce, larch appeared to encounter more internal competition for resources within organs. Indeed, we observed that the dates of the maximum rate occurrence were nearly the same between shoot growth and stem growth for larch. Shoots, however, represent <10% of the annual carbon allocation, whereas wood represents approximately 45% of the annual allocation, which is the largest carbon sink [37,38]. This coordination of growth in the shoot and stem could also suggest potential competition for resources use in time regardless of the low quantity of energy needed to produce new shoots. For spruce, in contrast, the staggered peaks observed within needle, shoot, and stem growth showed that spruce might guarantee a carbon supply and mediate resource competition by temporal asynchrony between organs [43]. If the future habitat conditions confine tree growth with predicted climate warming (e.g., evaporative demand increases), it may infer that the spruce will benefit more from declining water/carbon availability due to its high adaptability to acquire and allocate resources, and it could outcompete the larch at resourced-limited sites [22]. Consistent with this presumption, more spruce seedlings by natural regeneration than larch were scattered within the study site.

### 4.3. Environmental Effect on Growth

In our study, soil temperature was important for both primary growth and secondary growth (Table 4). Previous research in this study area has shown that soil temperature played a key role for the initiation of radial growth for both larch and spruce [8,44]. The threshold value of soil temperature for larch xylogenesis was above 0 °C as frozen soil will inhibit water uptake and activity in root systems [45]. Moreover, the correlation coefficients indicated that the growing degree days of soil temperature ($GDD_s$) better explained the shoot growth and stem growth, whereas the daily soil temperature was better for needle growth in larch (Table 4). In contrast, Antonucci et al. [7] reported that the GDD was more informative for apical meristems, while the thermal threshold better explained cambium phenology. This discrepancy between our findings and Antonucci's could be explained by different time ranges of growth because the latter only focused on the beginning, e.g., the onset of cambial cell division in the stem. In cold alpine ecotones, the cumulative temperature sum rather than the threshold temperature determines the duration of tree growth [46], which might be an adaptive response of the tree to avoid unfavorable conditions.

Within the two species, larch showed higher coefficients for environmental–growth relationships than spruce, especially the correlation between air temperature and growth. This finding is consistent with that of Oberhuber et al. [22], who pointed out that as a pioneer species, larch is light demanding and can grow rapidly at the early stage of forest succession, while as a late-successional species, evergreen spruce is a moderately shade-tolerant tree in the Alps. Pioneer species are often more sensitive to temperature and photoperiod insensitive once their chilling demand has been fulfilled, allowing them to grow as soon as the climate is favorable. Late-successional conifers, however, are often temperature insensitive and controlled by the photoperiod in spring once the critical day length has come, preventing them from beginning too early, as proposed by Cuny et al. [25].

Stem growth of the two species peaked at the same time and occurred in early July (Figure 6), which implies that extrinsic factors (e.g., climatic factors) not intrinsic factors (genetic factors) regulate the maximum rate of xylem growth [47]. The fact that this date occurred a few days after the summer solstice confirms the statement from Rossi et al. [48] that the photoperiod is the main factor controlling the timing of the fastest growth rate in boreal and temperate regions. This safety control may allow trees to complete cell wall lignifications within a limited growth period before the coming winter (cambial activity related to tree size).

## 5. Conclusions

In this work, we monitored the primary and secondary growth of larch and spruce trees on a weekly basis in North-Central China. Our results showed that primary growth preceded secondary growth in an onset sequence of needle, shoot, and stem without species-specific differences. This finding revealed that secondary growth might be dependent on auxin and/or carbon from newly developing leaves and shoots in the early growing season. Primary growth between the two species was asynchronous, with larch beginning budburst approximately 20 days earlier than spruce, which could be explained by the different life forms of their leaves (deciduous versus evergreen). However, secondary growth between larch and spruce were synchronic, indicating intra-species competition for resources existed in the cold alpine treeline environment. Compared with spruce, larch appeared to encounter more internal competition of resources within organs because the maximum growth rate of the shoot and stem occurred at the same time for larch but was staggered for spruce. We foresee that spruce will benefit more due to its high adaptability to acquiring and allocating resources, and it could outcompete larch at resourced-limited sites. These results provide insight into the potential physiological correlation between primary and secondary growth and assist our understanding of different growth responses of co-occurring coniferous trees under the cold alpine treeline, which will certainly help us assess the potential effects of climate change on tree growth, predict forest stand structure dynamics, and quantify forest ecosystem productivity.

**Supplementary Materials:** The following are available online at http://www.mdpi.com/1999-4907/10/7/574/s1, S1: GDD calculation, Table S1: Timing and duration of primary phenology (needle and shoot growth) of *Larix principis-rupprechtii* and *Picea meyeri*, Table S2: Timing and duration of secondary phenology (stem cambial activity and xylem differentiation) of *Larix principis-rupprechtii* and *Picea meyeri*, Table S3: Parameters (*A*, $t_p$, *k*) of the Gompertz functions fitted to the growth of needle and shoot length (mm) and stem xylem cell formation (cell number).

**Author Contributions:** Conceptualization, Y.J.; writing—original draft preparation, Y.Z.; investigation, Y.W., X.D. and B.W.; writing—review and editing, J.X.

**Funding:** This work is funded by the National Key Research and Development Program of China [Grant No. 2018YFA0606101], the National Natural Science Foundation of China (Grant No. 41630750 and 41801026), the China Scholarship Council (Grant No. 201808410575 and 201908410061) and the Starting Foundation for Doctors of Henan University of Science and Technology (No. 4026-13480057).

**Acknowledgments:** Special thanks to the anonymous referees for their valuable comments and suggestions.

**Conflicts of Interest:** The authors declare no conflict of interest.

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
