# Peer review of "Comparing Primary and Secondary Growth of Co-Occurring Deciduous and Evergreen Conifers in an Alpine Habitat"

_forests, doi:10.3390/f10070574_

Reviewer 1 Report

This is an excellent study, with a high effort to give a full portrait of the growth cycle of two contrasted conifers. I have enjoyed when reading the paper, as the comparison between a deciduous and an evergreen is a great choice to understand mechanisms related to growth rhythms.

Introduction is well written, in terms of the theoretical background. The hypothesis is well stated. The methodology is very adequately described and results well presented, Discussion follows the results wothout excessive "interpretation" of the data. So, I can only recommend the publication of this paper.

To be honest, I miss the comparison in - at least - two consecutive years to analyse the covariance effects of inetrannual changes. I did it when study a simmilar process and some interesting conclusiosn could be derived.

I recommend to edit the english. I am not a native english speaker but I have been able of finding some minor mistakes, specially in the abstract. Moreover, check lines 243-244...some error is evident.

Author Response

Cover Letter

Manuscript ID: foests-534188

Title: Comparing Primary and Secondary Growth of Co-occurring Deciduous and Evergreen Conifers in an alpine Habitat

 Dear Editor and Reviewers,

Thank you very much for giving us the opportunity to revise our manuscript. We do appreciate the professional comments and suggestions from the reviewers.

We must admit that only one-year research may miss some interesting inter-annual comparison. However, limited by the number of workman and fund supporting in those days, the sampling was conducted only one year and this is our flaw which should be improved in the future study.

According to the detailed suggestions from three reviewers, we have tried our best to revise our manuscript. More detail as below:

(1)   In Material and Methods part, we added a new Figure in chapter 2.1 Study site and species descriptions to visually clarify the weather condition during 2014. Meanwhile, we revised chapter 2.4 Weather station and included more information on what equipment was used in the weather station. Moreover, we reorganized chapter 2.2 Primary growth and added the information on the way how tree growth was measured.

(2)   In Result part, we revised Figure 6 and added original data points, which improve to describe the goodness of model fit.

(3)   Considering the research duration was only one year, we revised the term “climate” and modified it into “environment”.

(4)   We explained all the abbreviations, revised all the references and carefully check the whole manuscript.

(5)   Additionally, we have submitted our manuscript to the professional language company (Elsevier) for English language editing.

At last, we have carefully answered our response point-by-point to reviewers’ comments as follows, and have resubmitted to the new version which marked in highlighted red. We hope that this revision could produce an acceptable manuscript. Please kindly have a check.

Best regards.

Yours sincerely,

Yuan Jiang & Yiping Zhang

2019-06-27

 Response to Reviewer 1

This is an excellent study, with a high effort to give a full portrait of the growth cycle of two contrasted conifers. I have enjoyed when reading the paper, as the comparison between a deciduous and an evergreen is a great choice to understand mechanisms related to growth rhythms.

Introduction is well written, in terms of the theoretical background. The hypothesis is well stated. The methodology is very adequately described and results well presented, Discussion follows the results wothout excessive "interpretation" of the data. So, I can only recommend the publication of this paper.

Point 1: To be honest, I miss the comparison in - at least - two consecutive years to analyse the covariance effects of inetrannual changes. I did it when study a simmilar process and some interesting conclusiosn could be derived.

Response:

Thanks for your comment and suggestion. We must admit that only one-year research may miss some interesting inter-annual comparison. However, limited by the number of workman and fund supporting in those days, the sampling was conducted only one year and this is our flaw which should be improved in the future study.

Point 2: I recommend to edit the english. I am not a native english speaker but I have been able of finding some minor mistakes, specially in the abstract. Moreover, check lines 243-244...some error is evident.

Response:

 Thanks for your doubt and suggestion. We have accepted your advices and revised Abstract part. Moreover, we have checked L243-245 and deleted these unrelated sentences. Please see L255.

Reviewer 2 Report

In general, this manuscript is well-organized and demonstrates thoughtful analysis and content.  Modeling and comparing primary and secondary growth characteristics and phenology in deciduous and evergreen conifers has merit and is likely of interest to readers of Forests, and the authors included all of the necessary tests and comparisons.  The connection to climate change is fairly well argued, and the English is generally very good.  

However, there are limitations that greatly reduce the value of this manuscript in extending current knowledge.  The sample size is very small (5 trees per species), and the study was conducted in only one year, which was admittedly unusually wet.  Because there is high variability in growth and development in conifers, it is difficult to trust the analysis when taken from just 5 trees over one season.  Also, bud development was apparently tracked only on four buds per tree.  Much of the data were non-normal, which is not surprising with this type of data, but it is likely that the small sample size compounded the difficulty in analysis.   

A few recommendations: I think this study needs to be repeated, ideally in one or more years where weather differs from the original year, and using more trees to increase sample size.  A larger sample size may allow the authors to fit the data to an appropriate distribution or to transform the data to use the more powerful parametric tests.  Because variability is tree-growth is generally quite large, Fig.5 should include some indication of goodness of fit.  I have no way of knowing whether the Gompertz function was a 99% fit to the data, or 3%.  I generally include the original data points in the figure around the model line to demonstrate goodness of fit visually, and use some appropriate parameters to describe fit numerically depending on the model (e.g., R2).  In line 159ff, the authors should include information on what equipment was used in the weather station, where sensors were placed, etc. Was air temperature measured at 1 m above the ground? Was the sensor shielded from the sun?  What was the depth of the soil sensors?  What sensors were used?  All of these details are missing.  I would also prefer to see more discussion of GDD.  Accumulation of thermal time is well known to drive phenological development, and is generally a more useful measure than day-of-year.   

This project could provide valuable information and insight to help predict tree responses to climate change.  The study appears to be well-thought out and well-designed, but needs to be expanded.  I would encourage the authors to expand this study.

Author Response

Cover Letter

Manuscript ID: foests-534188

Title: Comparing Primary and Secondary Growth of Co-occurring Deciduous and Evergreen Conifers in an alpine Habitat

Dear Editor and Reviewers,

Thank you very much for giving us the opportunity to revise our manuscript. We do appreciate the professional comments and suggestions from the reviewers.

We must admit that only one-year research may miss some interesting inter-annual comparison. However, limited by the number of workman and fund supporting in those days, the sampling was conducted only one year and this is our flaw which should be improved in the future study.

According to the detailed suggestions from three reviewers, we have tried our best to revise our manuscript. More detail as below:

(1)   In Material and Methods part, we added a new Figure in chapter 2.1 Study site and species descriptions to visually clarify the weather condition during 2014. Meanwhile, we revised chapter 2.4 Weather station and included more information on what equipment was used in the weather station. Moreover, we reorganized chapter 2.2 Primary growth and added the information on the way how tree growth was measured.

(2)   In Result part, we revised Figure 6 and added original data points, which improve to describe the goodness of model fit.

(3)   Considering the research duration was only one year, we revised the term “climate” and modified it into “environment”.

(4)   We explained all the abbreviations, revised all the references and carefully check the whole manuscript.

(5)   Additionally, we have submitted our manuscript to the professional language company (Elsevier) for English language editing.

At last, we have carefully answered our response point-by-point to reviewers’ comments as follows, and have resubmitted to the new version which marked in highlighted red. We hope that this revision could produce an acceptable manuscript. Please kindly have a check.

Best regards.

Yours sincerely,

Yuan Jiang & Yiping Zhang

2019-06-27

 Response to Reviewer 2

Comments and Suggestions for Authors

In general, this manuscript is well-organized and demonstrates thoughtful analysis and content.  Modeling and comparing primary and secondary growth characteristics and phenology in deciduous and evergreen conifers has merit and is likely of interest to readers of Forests, and the authors included all of the necessary tests and comparisons.  The connection to climate change is fairly well argued, and the English is generally very good. 

 Point 1: However, there are limitations that greatly reduce the value of this manuscript in extending current knowledge.  The sample size is very small (5 trees per species), and the study was conducted in only one year, which was admittedly unusually wet.  Because there is high variability in growth and development in conifers, it is difficult to trust the analysis when taken from just 5 trees over one season.  Also, bud development was apparently tracked only on four buds per tree.  Much of the data were non-normal, which is not surprising with this type of data, but it is likely that the small sample size compounded the difficulty in analysis.  

 A few recommendations: I think this study needs to be repeated, ideally in one or more years where weather differs from the original year, and using more trees to increase sample size.  A larger sample size may allow the authors to fit the data to an appropriate distribution or to transform the data to use the more powerful parametric tests. Because variability is tree-growth is generally quite large.

Response:

Thanks for your comment and suggestion. We must admit that only one-year research may miss some interesting inter-annual comparison. Obviously, a longer sampling period and a larger sample size could improve data quality, build up more appropriate way to analyze data and produce more convincing result. However, limited by the number of workman and fund supporting in those days, the sampling was conducted only one year and this is our flaw which should be improved in the future study.

Point 2: Fig.5 should include some indication of goodness of fit.  I have no way of knowing whether the Gompertz function was a 99% fit to the data, or 3%.  I generally include the original data points in the figure around the model line to demonstrate goodness of fit visually, and use some appropriate parameters to describe fit numerically depending on the model (e.g., R2). 

Response:

 Thanks for your doubt and suggestion. We have accepted your advices and revised Figure 6 which contains original measured data to demonstrate the goodness of fit more visually and clearly. Besides, more Gompertz function model parameters had been given in Table S3 to describe fit numerically. Please see L230, 533.

 Point 3: In line 159ff, the authors should include information on what equipment was used in the weather station, where sensors were placed, etc. Was air temperature measured at 1 m above the ground? Was the sensor shielded from the sun?  What was the depth of the soil sensors?  What sensors were used?  All of these details are missing. 

Response:

 Thanks for your doubt and suggestion. We have accepted your advices and added the related information as below " In the study site (2740 m a.s.l.), an automatic weather station was installed in a relatively flat and unshielded area to measure environmental factors. The air temperature probe (HOBO Pro V2, Onset Computer Corp., USA) was fixed at a height of 1.5 m above the ground and the soil temperature probe (ST-05, Delta-T, UK) was buried at a depth of 10–20 cm. The soil volumetric water content (SWC) was observed (with a PR2 instrument, DeltaT, UK) at depth of 10–20 cm. Precipitation was also measured (with Davis 7852 instrument, Hayward, USA). All these environmental data were recorded at 30 min intervals and stored by CR10X data loggers (Campbell Scientific Corporation, Logan, UT, USA).". Please see L168-175.

 Point 4: I would also prefer to see more discussion of GDD.  Accumulation of thermal time is well known to drive phenological development, and is generally a more useful measure than day-of-year.  

Response:

 Thanks for your doubt and suggestion. We totally agree with your opinion on the importance of accumulated heat sum for phenological development and please see the discussion as below "Moreover, correlation coefficients indicated that growing degree days of soil temperature (GDDs) better explained the shoot growth and stem growth, whereas daily soil temperature was better for needle growth in larch (Table 4). In contrast, Antonucci et al. [7] reported that the GDD was more informative for apical meristems, while the thermal threshold better explained cambium phenology. This discrepancy between our findings and Antonucci’s could be explained by different time ranges of growth because the latter only focused on the beginning, e.g., the onset of cambial cell division in stem. In cold alpine ecotones, cumulative temperature sum rather than threshold temperature determines the duration of tree growth [46], which might be an adaptive response of tree to avoid unfavorable condition.". Please see L320-328.

 This project could provide valuable information and insight to help predict tree responses to climate change.  The study appears to be well-thought out and well-designed, but needs to be expanded.  I would encourage the authors to expand this study.

Reviewer 3 Report

The manuscript presents very interesting research combining analyses of primary and secondary growth of two different species in the treeline ecotone. The originality of the subject is even greater since coniferous and broadleaved species are compared.

The paper should be published after some revisions:

The general remarks.

The weakest point of the research is its duration. The measurements, sampling, and observations were performed only during one growing season. This almost disqualified the study from publication. The only way to accept the manuscript would be to add a very careful study of the weather during 2014 in comparison with the local climate. The one sentence (line 100) is absolutely not enough to cover this topic.

From the same reasons authors can not use the term climate when they talk about the weather.  Modify the text adequately.

The results of the correlations between primary and secondary growth weren't debated and compared with other studies. Include a paragraph about that in the discussion. Additionally, it would be beneficial to present the relations from Table 3 as a scatter graph.

other remarks:

1)  Although the authors mentioned shoots in chapter 2.2 (line 124) the details about way how the growth was measured are lacking. Add this information.

2) Nothing like "standard weather station" exists (line 160).  The parameters of the measurements have to be provided.

3) Explain way the Gompertz function (line 178) was selected to represents the trends of intraannual growth.

4) All abbreviations has to be explained.

Author Response

Cover Letter

Manuscript ID: foests-534188

Title: Comparing Primary and Secondary Growth of Co-occurring Deciduous and Evergreen Conifers in an alpine Habitat

 Dear Editor and Reviewers,

Thank you very much for giving us the opportunity to revise our manuscript. We do appreciate the professional comments and suggestions from the reviewers.

We must admit that only one-year research may miss some interesting inter-annual comparison. However, limited by the number of workman and fund supporting in those days, the sampling was conducted only one year and this is our flaw which should be improved in the future study.

According to the detailed suggestions from three reviewers, we have tried our best to revise our manuscript. More detail as below:

(1)   In Material and Methods part, we added a new Figure in chapter 2.1 Study site and species descriptions to visually clarify the weather condition during 2014. Meanwhile, we revised chapter 2.4 Weather station and included more information on what equipment was used in the weather station. Moreover, we reorganized chapter 2.2 Primary growth and added the information on the way how tree growth was measured.

(2)   In Result part, we revised Figure 6 and added original data points, which improve to describe the goodness of model fit.

(3)   Considering the research duration was only one year, we revised the term “climate” and modified it into “environment”.

(4)   We explained all the abbreviations, revised all the references and carefully check the whole manuscript.

(5)   Additionally, we have submitted our manuscript to the professional language company (Elsevier) for English language editing.

At last, we have carefully answered our response point-by-point to reviewers’ comments as follows, and have resubmitted to the new version which marked in highlighted red. We hope that this revision could produce an acceptable manuscript. Please kindly have a check.

Best regards.

Yours sincerely,

Yuan Jiang & Yiping Zhang

2019-06-27

 Response to Reviewer 3

Comments and Suggestions for Authors

The manuscript presents very interesting research combining analyses of primary and secondary growth of two different species in the treeline ecotone. The originality of the subject is even greater since coniferous and broadleaved species are compared.

 The paper should be published after some revisions:

The general remarks.

Point 1: The weakest point of the research is its duration. The measurements, sampling, and observations were performed only during one growing season. This almost disqualified the study from publication. The only way to accept the manuscript would be to add a very careful study of the weather during 2014 in comparison with the local climate. The one sentence (line 100) is absolutely not enough to cover this topic.

Response:

 Thanks for your doubt and suggestion. We have accepted your advices and added new Figure 2 to clarify more detail information of the weather in sampling year. Please see Figure 2. Additionally, 2014 was a wet year, however, these surplus precipitations mainly fell in July but not in the beginning of the growing season, which may limitedly affect tree early spring phenology. Please see L107-111.

Point 2: From the same reasons authors can not use the term climate when they talk about the weather.  Modify the text adequately.

Response:

 Thanks for your doubt. We agreed the opinion that using the term “climate” was unsuitable due to too short study period. Considering some environmental factors were also included in analysis (e.g. soil temperature and soil water content), we have modified the term into “environment”. Please see L87, 242, 315, 329.

 Point 3: The results of the correlations between primary and secondary growth weren't debated and compared with other studies. Include a paragraph about that in the discussion.

Response:

     Thanks for your comment. Please see this content in Results part 3.3. Comparing primary and secondary growth and Discussion part 4.1. Coordination and correlation between primary and secondary growth.

 other remarks:

Point 4: Although the authors mentioned shoots in chapter 2.2 (line 124) the details about way how the growth was measured are lacking. Add this information.

Response:

    Thanks! We have taken the suggestions and added revised them as below "shoot elongation was determined on long-shoots for both larch and spruce. …The length of needle and shoot elongation were measured by a steel ruler with 1 mm precision.". Please see L131-132, 135-136.

 Point 5: Nothing like "standard weather station" exists (line 160).  The parameters of the measurements have to be provided.

Response:

    Thanks! We have taken the suggestions and revised the description on chapter 2.4 weather station. Moreover, we have added more detail information of monitored sensors. Please see L168-175.

 Point 6: Explain way the Gompertz function (line 178) was selected to represents the trends of intraannual growth.

Response:

 Thanks for your comment. Please see the explanation in L185-186.

 Point 7: All abbreviations has to be explained.

Response:

 Thanks for your comment and suggestion. We have carefully checked the abbreviations in manuscript thoroughly to make sure all of them had been explained.

Round  2

Reviewer 2 Report

As revised, this manuscript is well-structured and I feel it would make a valuable contribution to the literature and is suitable for publication.  I unfortunately do not have the time to reread this in detail, but will assume the English grammar errors have been corrected, with the exception of the following that I noted in the text newly added by the authors in red.  Please make these changes:

 Line 102 should read: “However, this surplus precipitation mainly fell in July as 192.6 mm of rain, which was 70.4% higher than the long-term average for the same period (1957–2013, Figure 2).”

 Line 172 please add the word “a” before “depth” as: “The soil volumetric water content (SWC) was observed (with a PR2 instrument, DeltaT, UK) at a depth of 10–20 cm.”

 In line 174, if there is only one data logger in the weather station (as is usual), please change to “were recorded at 30 min intervals by a CR10X datalogger”

 Please make two changes to line 328 as: “which might be an adaptive response of the tree to avoid unfavorable conditions.”

 The content of the newly added figures appears to be good, but the print quality of the figures is inadequate.  Please upload high quality (high dpi) images (unless the problem is only with what was sent to reviewers).

 Otherwise, I am happy with the changes made by the authors and the explanations given, and recommend the manuscript for publication. 

Author Response

Cover Letter

Manuscript ID: foests-534188

Title: Comparing Primary and Secondary Growth of Co-occurring Deciduous and Evergreen Conifers in an alpine Habitat

 Dear Editor and Reviewers,

We are most appreciative for your kind work and constructive comments on our paper. We have studied the reviewer’ comments carefully and have submitted the new version based on the advices (marked in highlighted red).

Moreover, we have uploaded all figures with high quality for further editing. We hope that this revision could produce an acceptable manuscript. Please kindly have a check.

Best regards.

Yours sincerely,

Yuan Jiang & Yiping Zhang

2019-07-03

 Comments and Suggestions for Authors

As revised, this manuscript is well-structured and I feel it would make a valuable contribution to the literature and is suitable for publication.  I unfortunately do not have the time to reread this in detail, but will assume the English grammar errors have been corrected, with the exception of the following that I noted in the text newly added by the authors in red.  Please make these changes:

 Line 102 should read: “However, this surplus precipitation mainly fell in July as 192.6 mm of rain, which was 70.4% higher than the long-term average for the same period (1957–2013, Figure 2).”

Line 172 please add the word “a” before “depth” as: “The soil volumetric water content (SWC) was observed (with a PR2 instrument, DeltaT, UK) at a depth of 10–20 cm.”

In line 174, if there is only one data logger in the weather station (as is usual), please change to “were recorded at 30 min intervals by a CR10X datalogger”

Please make two changes to line 328 as: “which might be an adaptive response of the tree to avoid unfavorable conditions.”

Response:

Thanks! We have fully accepted the suggestion and modified these sentences. Please see in L102-103, 172, 174 and 328. 

 The content of the newly added figures appears to be good, but the print quality of the figures is inadequate.  Please upload high quality (high dpi) images (unless the problem is only with what was sent to reviewers).

Response:

Thank you for your comment and suggestion! We have uploaded all the figures with high quality. 

  Otherwise, I am happy with the changes made by the authors and the explanations given, and recommend the manuscript for publication. 

Reviewer 3 Report

The authors have made most of the required revisions and I believe manuscript can be published in the current state.

Author Response

Cover Letter

Manuscript ID: foests-534188

Title: Comparing Primary and Secondary Growth of Co-occurring Deciduous and Evergreen Conifers in an alpine Habitat

 Dear Editor and Reviewers,

We are most appreciative for your kind work and constructive comments on our paper. We have studied the reviewer’ comments carefully and have submitted the new version based on the advices (marked in highlighted red).

Moreover, we have uploaded all figures with high quality for further editing. We hope that this revision could produce an acceptable manuscript. Please kindly have a check.

Best regards.

Yours sincerely,

Yuan Jiang & Yiping Zhang

2019-07-03